# Differences in access to water, sanitation, and hygiene facilities among residents of Korail Slum, Bangladesh, during normal vs. water-logging situations

Md Mostafizur Rahman[1,2], Virasakdi Chongsuvivatwong[1], Alan F. Geater[1], Md Shamim Hayder Talukdar[2], Wit Wichaidit[1]*

1 Department of Epidemiology, Faculty of Medicine, Prince of Songkla University, Hat Yai, Thailand,
2 Eminence Associates for Social Development, Dhaka, Bangladesh

* wit.w@psu.ac.th

## Abstract

### Background

Slum areas in Dhaka, Bangladesh, experience frequent water-logging after heavy rain. However, the extent to which access to water, sanitation, and hygiene (WASH) varies between normal vs. water-logging periods has not been described. The objectives of this study are: 1) to describe WASH access and behaviors among Korail Slum residents during normal vs. water-logging periods, and; 2) to describe the extent to which the differences in WASH access and behaviors between normal vs. water-logging periods varied by socioeconomic status (SES).

### Methods

We conducted a cross-sectional study in Korail Slum, Dhaka, Bangladesh, during November 2024 using face-to-face interviews and rapid observation of WASH infrastructures. We selected adult residents to Korail Slum via systematic random sampling and invited them to participate. We interviewed participants regarding WASH access and behaviors, as well as their education, income, and asset ownership (i.e., socioeconomic status (SES) indicators). We categorized participants into SES tertiles using principal component analysis (PCA). We analyzed data using descriptive statistics and McNemar's tests.

### Results

Nearly all of our participants (n = 404) reported a Basic level of water access during normal and water-logging periods, and no participant reported using unimproved latrines or open defecation. Approximately 85% of the participants reported access to Basic hygiene facilities during normal periods, most (95%) of whom also reported

**Data availability statement:** All relevant data are within the paper and its Supporting Information files.

**Funding:** The first author (MMR) received financial support for data collection in this research work from the TUYF Charitable Trust: Research Capacity through Education and Networking on Epidemiology in Asia, the Department of Epidemiology, Faculty of Medicine, Prince of Songkla University (Grant number 1/2023). The funders had no role in study design, data collection and analysis, decision to publish, or preparation of the manuscript.

**Competing interests:** The authors have declared that no competing interests exist.

the access during water-logging periods (p-value = 0.134). The majority of participants did not always wash their hands at key moments during the normal period. Among those who reported always washing hands during the normal period, between 15% to 30% regressed and did not always wash their hands during the water-logging period. There were no statistically significant variations in the difference between WASH access and behaviors at normal vs. water-logging periods by socioeconomic tertile.

## Conclusion

We found near-universal access to water and improved sanitation, but low level of hand hygiene behaviors. Limitations regarding generalizability and potential social desirability bias should be considered as caveats in the interpretation of the study findings.

## Introduction

Bangladesh ranks as one of the most disaster-prone countries in the world, partly due to climate change [1]. Bangladesh experiences an average of two to five major floods per year, [2] which can inundate up to two-thirds of the country and cause severe damage to infrastructure, agriculture, and public health systems. However, to distinguish rural-area flooding in Bangladesh that forms a normal part of the agricultural cycle (particularly for growing rice) from the urban-area flooding that can affect the health of residents, the term "water-logging" will be used in this document in accordance with the literature [3] to refer to the accumulation of high-intensity rainfall runoff in urban areas when drainage is either hindered or not functional [3].

Dhaka, the capital of Bangladesh, is one of the most densely populated cities in South Asia. Dhaka faces extensive water-logging during the monsoon season (May to October) [4]. Over the past ten years, the frequency and intensity of flooding in Dhaka have increased [5]. Water-logging creates adverse social, physical, economic, environmental, and health problems [6], including typhoid, cholera, malaria, diarrhea, gastroenteritis, and dengue fever [7].

Water-logging disproportionately affects the health of the urban poor [7], who tend to aggregate in high-density slum areas. Approximately 70% of households in Dhaka slums are poorly built [3]. Slum residents are exposed to the open drainage system, particularly during water-logging. Korail Slum is the largest slum in Dhaka. Korail Slum residents are vulnerable to infectious diseases during water-logging [8–10]. Water, sanitation, and hygiene (WASH) have been shown to effectively prevent communicable diseases in urban slum settings [11]. However, water-logging can disrupt access to WASH facilities and indirectly affect health [12,13].

Although slum residents are generally less wealthy than residents of other parts of a city [14], heterogeneity in socioeconomic conditions (i.e., wealth disparities) can also be found within a slum [15] and can be measured using a combination of socioeconomic characteristics and asset ownership [16]. Despite concerns

regarding the vulnerability of slum residents to climate events and the general notion that climate events affect the poor more commonly and to a greater extent compared to the rich [17], studies have not described differences in access to WASH facilities within a household between normal and water-logging periods among slum residents. Studies also have not described the disparities of these differences by wealth. Such data can inform targeted interventions by helping public health and urban planning stakeholders identify population groups that are vulnerable to WASH disruptions during climate-related events. Therefore, the objectives of this study are: 1) to describe WASH access and behaviors among Korail Slum Residents during normal vs. water-logging periods, and 2) to describe the extent to which the differences in WASH access and behaviors between normal vs. water-logging periods varied by socioeconomic status (SES).

## Methods

### Study design and setting

We conducted a community-based cross-sectional study using structured interviews and non-participatory observations among households in Korail Slum, Dhaka, Bangladesh, during normal and water-logging situations. Data collection took place between 15–20 November 2024. Korail Slum is one of the largest informal settlements in Dhaka, spans approximately 100 acres in the middle of the city (Fig 1), and is home to around 16,500 households and 151,500 people [18]. The slum is characterized by overcrowded housing, narrow roads, and limited infrastructure. We selected the Korail Slum due to the geographic location in the middle of Dhaka which eases travel and access, the large population, and the vulnerability of the slum to water-logging.

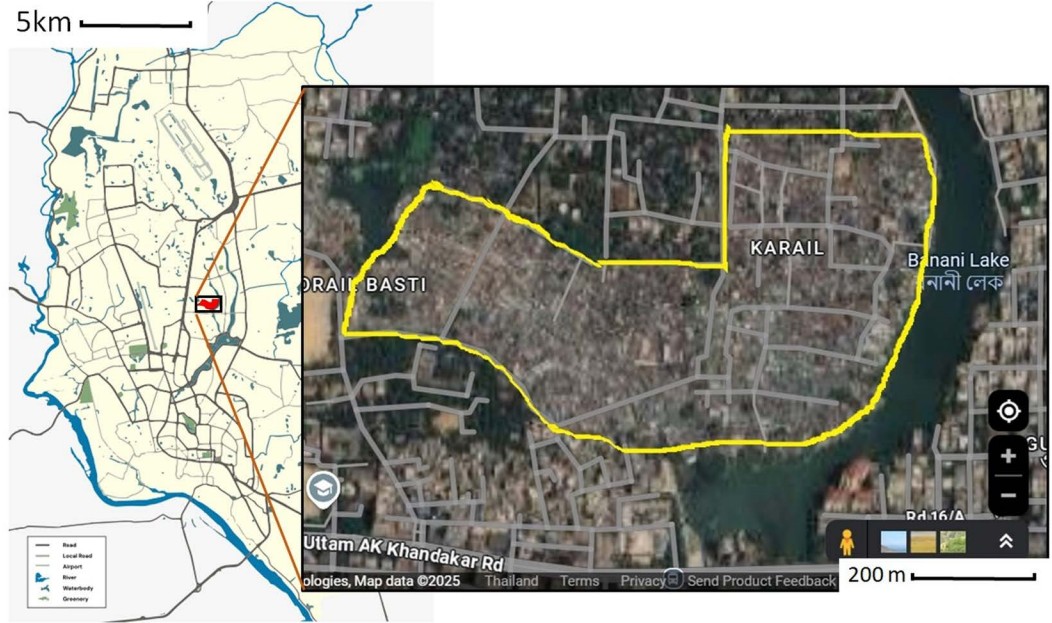

**Fig 1. Approximate outline of Korail Slum area within the city of Dhaka, Bangladesh – the yellow line designates the Slum's boundaries (image modified from Google Map).** Footnotes: Dhaka city map modified from an image titled Dhaka City Map 2025 by Ahnaf Tahmid Manan – Own work, CC BY-SA 4.0, https://commons.wikimedia.org/w/index.php?curid=163166332; Inset image with the Slum's boundaries modified from Google Map by the authors.

## Study participants and sample size calculation

The study population included adult residents of Korail with the following inclusion criteria: 1) Age 18 years or older; 2) Having resided in Korail for one year or longer; 3) Able to communicate in Bengali. We excluded individuals who self-reported illnesses or other impairments that they deemed would prevent them from participating fully in the interview.

We calculated the sample size for the survey based on the primary objective of describing the proportion of Korail Slum residents with access to WASH facilities. As no previous data on access to basic WASH facilities in Korail slum existed, we assumed that the proportion of the urban poor with access to WASH facilities in Korail Slums was the same as that in a previous study among resource-poor population in Bangladesh (27%, or p = 0.27) [19]. At 95% confidence and 5% arbitrary margin of error, we performed sample size calculation using the *epicalc* package in R [20] and obtained the sample size of 303 residents of Korail slum. We assumed that an arbitrary 25% of potential participants would refuse to participate in the study, thus we adjusted the sample size to 404 residents.

## Sampling method

We used systematic random sampling to select the households for the survey. The sampling frame for the study was derived from the Urban Health and Demographic Surveillance Systems (UHDSS) operated in selected slums of Dhaka and Gazipur, including Korail Slum. We used the 2024 version of the UHDSS, which included 16,536 households with contact information [18]. Our sample size was 404 residents, thus the sampling interval was 16536/404 ≈ 38 households. We approached the adult household members, provided information about the study, and invited the household member to participate. If a selected household had more than one eligible member, we used the "next birthday" method to identify the potential participant.

## Study instruments

We collected data by face-to-face interviews and rapid observation of WASH infrastructures in the participants' household premises. Our study instrument was a structured questionnaire hosted on the web-based KoboToolbox platform. We used smartphones with either a web browser or the KoboCollect application for data collection.

Initially, we developed an English questionnaire adapted from previous studies [21] and the WHO/UNICEF Joint Monitoring Programme (JMP) [22]. The lead author (MR) translated the questionnaire from English to Bengali, and another colleague in Bangladesh back-translated the questionnaire from Bengali to English. The lead author (MR) compared the original and the back-translated English versions, identified discrepancies, and made corrections to the corresponding parts in the Bengali translation. We subsequently finalized the draft questionnaire. We then pilot-tested the draft Bengali questionnaire in Mirpur Slum, Dhaka, to evaluate its phrasing, structure, length, and sequencing. We made revisions based on feedback from the pilot-testing session, and finalized the questionnaire for data collection.

## Study variables: Access to water, access to sanitation facility, and hygiene behaviors at water-logging and normal periods

In this study, the term "normal period" refers to the period in which the participant resided in Korail Slum but did not experience water-logging, and the term "water-logging period" refers to the period in which the participant resided in Korail Slum and experienced water-logging. Our operational definition of the term "*water-logging*" was "*inundation of at least 25 cm in depth lasting 10 hours or more*." We trained our research assistants to explain this definition in a uniform manner to all participants when conducting interviews about experiences during water-logging periods. We asked participants to describe their level of access to WASH during the most recent water-logging periods within the past 12 months that the participants experienced without giving the specific dates.

**Access to water during normal periods.** Investigators assessed access to water during normal periods through structured interviews and observations of water sources. Participants were asked to identify their main source of drinking water (e.g., piped water, tubewell, public tap, dug well, water from spring, rainwater, delivered water, water kiosk, packaged water, surface water) and the main source of water for other household activities (e.g., cooking, handwashing). Additionally, questions covered time spent collecting water, availability of water, queuing time, water management, payment for water use, and frequency of water shortages. During rapid observations, investigators asked participants to show their current drinking water containers and noted whether the containers were covered or uncovered and noted the observable method of drinking water treatment (e.g., boiling, filtering, chlorination).

**Access to water during water-logging periods.** Investigators assessed access to water during water-logging periods by asking participants whether they continued using the same main source of drinking water as in normal periods or switched to a different source due to water-logging. If a participant switched to a different source during water logging, we would ask the participant to identify the source using the same answer choices as our measurement of access to water during normal periods. We also asked the participants about changes in collection time, water shortages, and treatment practices. During rapid observations, investigators asked participants to show the source of drinking water during the most recent water-logging period. Investigators noted whether the containers were covered or uncovered and noted the observable method of drinking water treatment (e.g., boiling, filtering, chlorination).

**Access to sanitation facility during normal periods.** Investigators measured access to sanitation during normal periods by asking participants about the type of toilet facility currently used by household members (e.g., flush/pour flush toilets, dry pit latrines, composting toilets, bucket, container-based sanitation, hanging toilet/bush/field). Investigators also asked whether the facility was private, shared, or public, and the location of the toilet (in own dwelling, in own block/plot, or elsewhere). During rapid observations, investigators asked participants to show the toilet currently used by the participant and their household members, then noted the type of toilet facility and whether there was a clear path to the toilet.

**Access to sanitation facility water-logging periods.** Investigators asked participants about the type of toilet used during the latest water-logging period and whether the participants needed to change the toilet facility. During rapid observations, investigators asked participants to show the toilet used by the participant and their household members during the more recent water-logging period and noted the type of toilet facility and whether there was a clear path to the toilet.

**Hand hygiene facility access and behaviors during normal periods.** Investigators measured access to hand hygiene facility based on rapid observation. Investigators asked participants to show where the participants and their household members most frequently washed their hands, and observed the availability of water and soap. Investigators also asked the participants to self-report the frequency of handwashing with water and soap at key moments for hand hygiene. Key moments for hand hygiene referred to events identified by the WHO/UNICEF Joint Monitoring Programme [23] as moments where hand hygiene is essential to prevent disease transmission, namely: after going to the toilet, after changing a child's soiled diaper, before preparing food, before feeding a child, and before eating.

**Hand hygiene facility access and behaviors during water-logging periods.** Investigators asked participants to show where the participants and their household members most frequently washed their hands during the most recent water-logging period and observed the availability of water and soap. Investigators also asked the participants to self-report the frequency of handwashing with water and soap at key moments for hand hygiene.

## Measurement of socioeconomic status

We included questions regarding the participants' education, income, and asset ownership to measure socioeconomic status among the study participants. The investigators dichotomized education into two categories (secondary education or higher vs. primary education or less), income (more than 20,000 BDT per month vs. 20,000 BDT or less), and

availability of various household items. The household income of 20,000 BDT per month was approximately half of the average household income in urban areas of Bangladesh in 2022 [24], and we used the amount as a proxy cut-off point for being low-income in the slum setting. We then retained characteristics and assets with adequate heterogeneity of the distribution (i.e., more than 10% or less than 90% of all participants reported having the characteristic or asset) for Principal Component Analysis. The investigators then followed the instructions for Principal Component Analysis (PCA) [25] and obtained the factor scores. We then performed descriptive statistical analysis on the factor scores and used the 33rd and 67th percentiles as the cut-off points for the socioeconomic tertiles.

## Data collection

We recruited research assistants for this study from social and health sciences students and graduates from local universities. The research assistants underwent three days of training on research protocols and interview techniques, and a field practice session in an area outside Korail Slum with similar demographic and socioeconomic characteristics.

After receiving ethical approval from the Bangladesh Medical Research Council (BMRC), we sought permission from community leaders, then we asked for assistance from local community health workers to escort the research assistants to the study households. Before recruiting each participant, the research assistants would inform the participant about the study's purpose and procedures and ensure understanding regarding the voluntary nature of participation and confidentiality. We asked for verbal informed consent in lieu of written informed consent in order to accommodate to the context of Korail Slum where literacy level among the residents remained low.

After receiving verbal informed consent, research assistants then started the face-to-face interviews. After completing the face-to-face interview, research assistants then asked the participant for permission to observe WASH facilities used by the participant and their household members. All data were securely stored in an encrypted system with password protection to ensure confidentiality and data security. Data collection took place between 15–20 November 2024.

## Data management

We did not enter any personally identifiable data into the system to ensure the confidentiality of the study participants. We conducted regular checks to identify and resolve data-related issues (e.g., missing values, duplicated entries). We also performed routine data cleaning and prepared the final dataset for statistical analysis.

## Data analysis

We analyzed the study data using descriptive statistics to summarize self-reported hand hygiene behaviors and observed household wash conditions. Additionally, we performed cross-tabulations to compare self-reported handwashing behaviors with the observed availability of water and soap at household locations. In the cross-tabulation, we excluded all households where the participants reported no experience of water-logging within 12 months prior to the survey. We performed all statistical analyses using R software.

## Human research ethics

We requested and received ethical approval for this study from the Bangladesh Medical Research Council (BMRC) (Ref: BMRC/NREC/2022–2025/561). Due to the low literacy levels among the residents of Korail Slum, we requested and received approval for a waiver of written informed consent and the use of only verbal informed consent.

We obtained verbal informed consent from all participants before initiating data collection. This process included reading a consent script aloud. The consent script included a summary of the purpose of the study, procedures, risks, benefits, and confidentiality. We gave the participants an opportunity to ask questions before verbally agreeing to participate or declining participation. Our research assistants documented the verbal consent directly in the electronic questionnaire

form on KoboCollect. The form required an affirmative response regarding verbal consent to display the data collection questionnaire. During the information and informed consent process, our research assistants periodically reminded the participants that they were allowed to stop the interview or ask the investigation team to stop observing household conditions at any time. We instructed the research assistants to stop data collection (i.e., withdraw study participants) in case of emergency or when they deemed data collection to disrupt necessary activities within the household.

## Results

Our study included 404 participants (Table 1) with no refusal to participate (participation = 100%). Most participants were female, with a mean age of 34.2 years, married, and worked as housewives. Nearly all participants were Muslim. Education levels varied, with one-sixth reported having no schooling and another one-sixth reported having completed secondary or higher secondary education. Approximately two-fifths of the participants reported that their household earned between 20,000 Bangladeshi Taka (BDT) per month or less. More than half of the participants reported that their households had between 3–5 members. There were 21 participants who did not report water-logging by the household within 12 months before the survey. Thus, the comparison of access between water-logging vs. non-logging periods included 383 participants.

Analyses of data regarding socioeconomic status and household asset ownership (S1 Table) showed that there was universal access to electricity, and nearly 90 percent of the participants reported having a smartphone. However, there was heterogeneity with regard to educational attainment, income, television ownership, refrigerator ownership, etc. Principal component analysis also showed substantial loading on these items (S2 Table). The investigators then used the principal component analysis output to rank participants into socioeconomic tertiles, and cross-tabulation of education, income, and asset ownership by tertile also showed consistency: participants in the third higher socioeconomic tertile were more likely than participants in the first tertile to have secondary education, to have more than 20,000 BDT per month of income, and to be in households with a television, a refrigenrator, etc. (S3 Table).

During normal periods, all participants reported access to basic water services, and only one participant reported a downgrade to limited service during water-logging period (Fig 2 & S4 Table). As only one participant reported limited instead of basic access to water, we found no variations in access to water by wealth tertiles.

Sanitation access remained similarly stable: most participants reported using shared improved latrines (hence at the Limited level) at both normal and water-logging periods (Fig 3 & S5 Table). There was no report of using unimproved latrine or open defecation during water-logging periods. Those in Tier 2 with a Basic level of access during normal periods were more likely to change from a Basic level during normal periods to a Limited level during water-logging. However, these differences by socioeconomic status were not statistically significant according to the Breslow-Day test.

Hand hygiene facility access varied markedly between normal vs. water-logging periods (Fig 4 & S6 Table). Over 95% of those who reported basic access to hygiene facilities (i.e., having both water and soap at the handwashing place) during normal period also reported basic access during the water-logging period. Among those with limited access to hygiene facilities (i.e., having only water at the handwashing place) during normal period, nearly one-fifth reported having no facility during water-logging. However, the differences were not statistically significant. Participants in the first tertile of wealth (poorer) were more likely to change from Basic during the normal period to No Facility during the water-logging period, compared to those in the second tertile (middle).

We also found differences in self-reported handwashing behaviors at key moments (Fig 4 & S6 Table). Nearly all of those who reported not always washing hands during normal period also reported not always washing hands during water-logging period. Among those who reported always washing hands during normal period, only 15% to 30% regressed and did not always wash their hands during water-logging period; these differences were statistically non-significant. Those who reported always washing hands after toilet use in the second wealth tertile were less likely than those in the first and third wealth tertiles to do so. Similar patterns were found for handwashing before preparing food, handwashing before feeding a child, and handwashing before eating.

**Table 1. Characteristics of the Study Participants and the Participant's Household (n = 404 Participants).**

| Characteristic | Frequency (%), unless otherwise indicated |
|---|---|
| **Sex, (n/%)** | |
| Male | 94 (23.0%) |
| Female | 310 (77.0%) |
| **Age (years), (mean ±SD)** | 34.20±11.63 |
| **Occupation, (n/%)** | |
| Housewife | 244 (60.4%) |
| Others[a] | 130 (32.2%) |
| Not in employment (Student, not working/disabled, or retired/homemaker) | 30 (7.4%) |
| **Religion** | |
| Islam | 403 (99.8%) |
| Buddhism | 1 (0.2%) |
| **Marital status, (n/%)** | |
| Single/Never married | 24 (5.9%) |
| Married | 372 (92%) |
| Divorced | 2 (0.5%) |
| Widowed/Separated | 6 (1.5%) |
| **Highest level of education completed** | |
| Never went to school | 64 (16.0%) |
| Primary school | 72 (18.0%) |
| Secondary education (did not obtain SSC) | 99 (24.5%) |
| SSC/equivalent (Year 10) | 70 (17.0%) |
| HSC/equivalent (Year 12) | 70 (17.5%) |
| Bachelor's degree or higher | 29 (7.1%) |
| **Household Monthly Income** | (n = 403 participants) |
| No more than 10,000 Taka | 28 (7.0%) |
| 10,001–20,000 Taka | 134 (33.3%) |
| 20,001–30,000 Taka | 135 (33.5%) |
| 30,001–40,000 Taka | 66 (16.4%) |
| More than 40,000 Taka | 40 (10.0%) |
| **Household members (number of persons), median (IQR)** | 4 (3, 5) |
| **Relation with the head of household** | |
| Respondent is the head of household | 97 (24.0%) |
| Wife/husband/partner | 250 (62.0%) |
| Others | 57 (14.0%) |

[a]Including Rickshaw/van/cart pullers, garments worker, construction workers, drivers, vendors, service workers, hawkers, porter/day laborers, and servants/maid servants

## Discussion

In this community-based cross-sectional study, we described differences in access to WASH facilities among Korail Slum residents during normal and water-logging periods, as well as differences in self-reported handwashing with water and soap during normal and water-logging periods. We found very few differences regarding the access to water and sanitation facilities during normal and water-logging periods. However, we found a pattern of change in self-reported hand

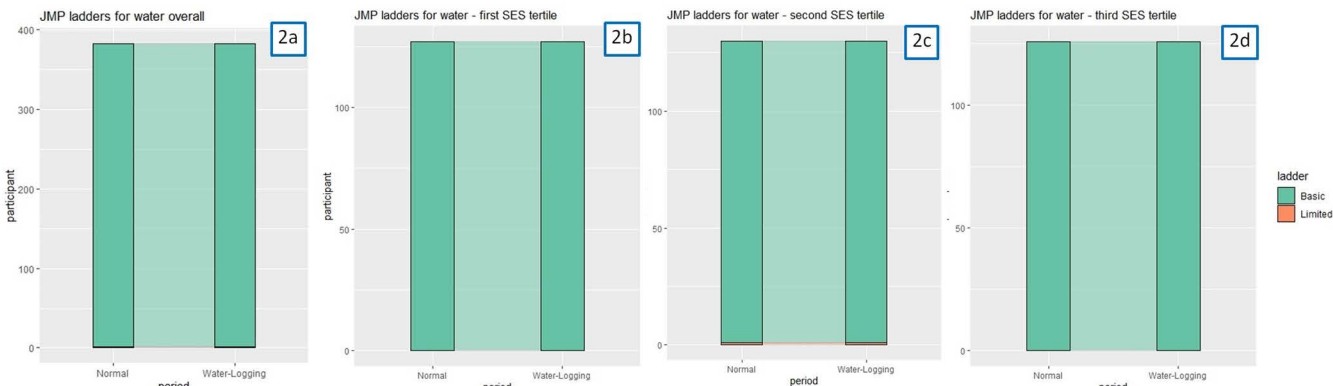

**Fig 2. Access to water according to the JMP Service Ladder among participating Korail Slum residents during normal vs. water-logging periods.** Footnotes: 2a) Overall; 2b) Among participants in the first SES tertile; 2c) Among participants in the second SES tertile; 2d) Among participants in the third SES tertile.

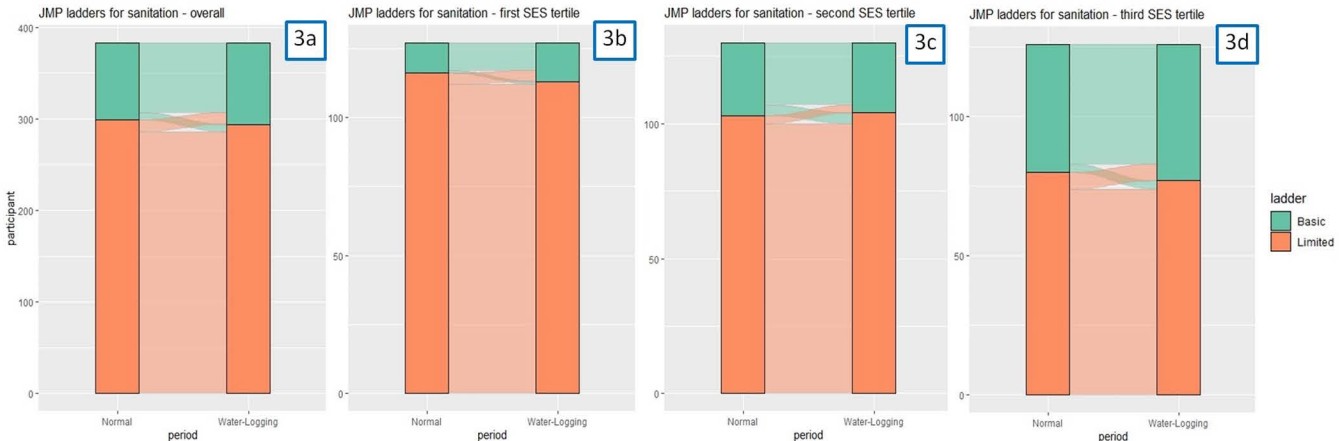

**Fig 3. Access to sanitation facility according to the JMP Service Ladder among participating Korail Slum residents during normal vs. water-logging periods.** Footnotes: 3a) Overall; 3b) Among participants in the first SES tertile; 3c) Among participants in the second SES tertile; 3d) Among participants in the third SES tertile.

hygiene behaviors. Those who did not always wash their hands during normal periods also did not always wash their hands during water-logging periods. However, some of those who always washed their hands during normal periods did not continue the behavior during water-logging periods. With regard to effect modification by socioeconomic status, we found that changes in self-reported handwashing with soap from "always" during normal periods to "not always" during water-logging periods were most common among those in the second (middle) tertile of wealth.

## Changes in WASH access and behaviours during water-logging

All except one of our participants reported having access to Basic water facility. In other words, in Korail slum, there was practically universal access to a water facility at the Basic level according to the Joint Monitoring Program (JMP) classification. Similarly, with regard to sanitation, no participant reported using unimproved latrines or engaging in open defecation. The absence of open defecation suggested that the prevalence of open defecation in Bangladesh continued to decline after the reduction to

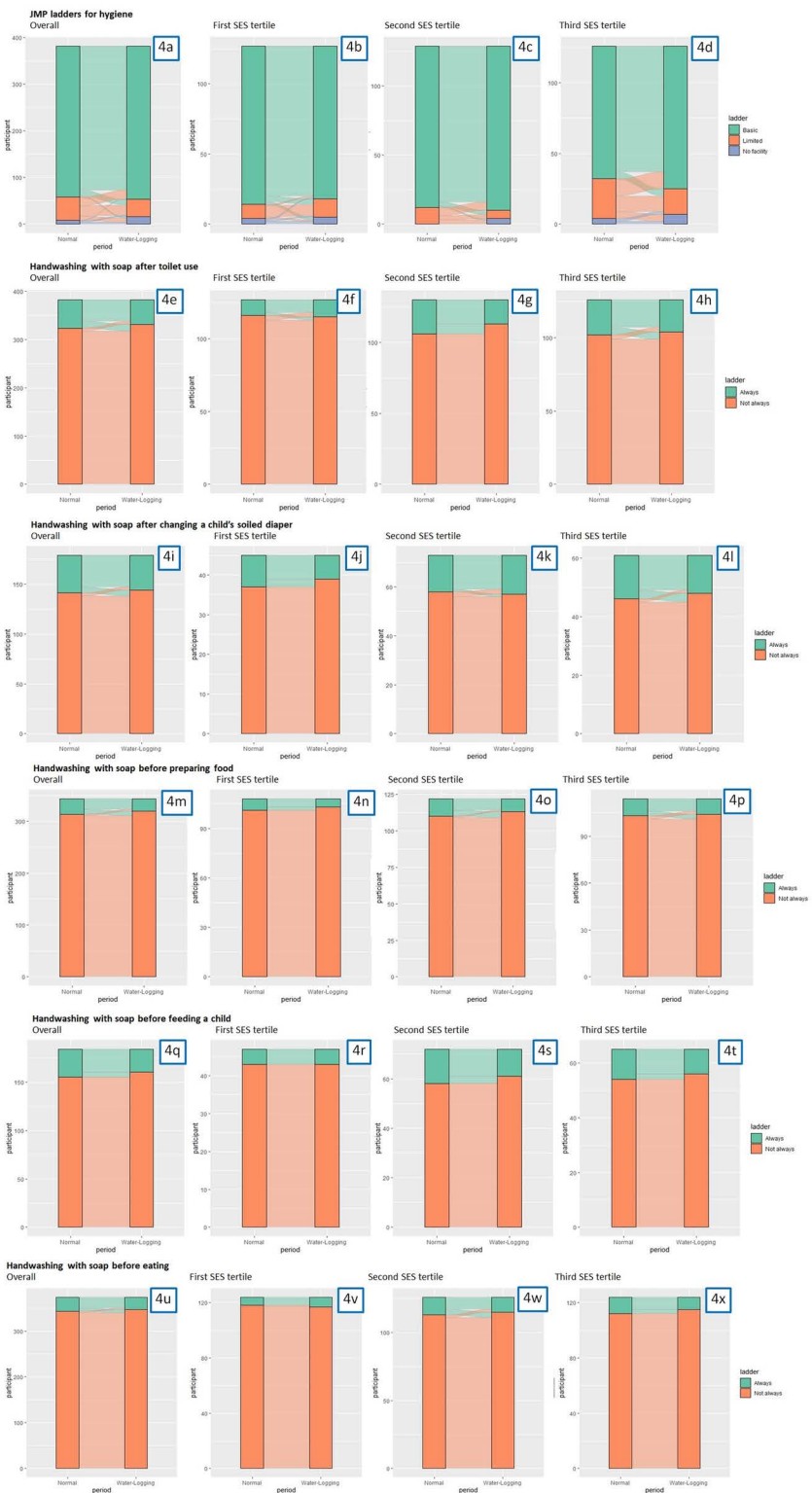

**Fig 4. Access to hygiene facility according to the JMP Service Ladder and handwashing with soap among participating Korail Slum residents during normal vs. water-logging periods.** Footnotes: 4a) Access to Hand Hygiene Facility Overall; 4b) Among participants in the first SES tertile; 4c) Among participants in the second SES tertile; 4d) Among participants in the third SES tertile. 4e) Handwashing with Soap After Toilet Use Overall; 4f)

Among participants in the first SES tertile; 4g) Among participants in the second SES tertile; 4h) Among participants in the third SES tertile. 4i) Hand-washing with Soap After Changing a Child's Soiled Diaper Overall; 4j) Among participants in the first SES tertile; 4k) Among participants in the second SES tertile; 4l) Among participants in the third SES tertile. 4m) Handwashing with Soap Before Preparing Food Overall; 4n) Among participants in the first SES tertile; 4o) Among participants in the second SES tertile; 4p) Among participants in the third SES tertile. 4q) Handwashing with Soap Before Feeding a Child Overall; 4r) Among participants in the first SES tertile; 4s) Among participants in the second SES tertile; 4t) Among participants in the third SES tertile. 4u) Handwashing with Soap Before Eating Overall; 4v) Among participants in the first SES tertile; 4w) Among participants in the second SES tertile; 4x) Among participants in the third SES tertile.

one percent in 2015 [26]. These major shifts are possibly attributable to significant investments in water and sanitation infrastructures and behavior change campaigns. However, the validity of this self-reported absence of open defecation warrants scrutiny. Social desirability bias may have influenced responses, as participants may have been embarrassed to admit engaging in open defecation after being exposed to widespread sanitation-focused awareness campaigns [27]. The majority of households had limited sanitation during normal periods due to the act of sharing their improved latrine with people from other households. However, a small minority (approximately four percent) of participants from these households reported having access to basic (non-shared) sanitation facilities during water-logging periods. We do not know what accounted for this shift, although a previous study reported that slum residents would relocate housing within the same area during water-logging [28], such as with family members, and relocated residence might have non-shared toilet. Future studies should consider asking questions regarding temporary relocation during water-logging periods. One additional issue in our data is that we did not collect information regarding water and sanitation related behaviors (e.g., the frequency of drawing water or using sanitation facilities). Future studies should consider including behavioral information in order to understand water and sanitation access in a more complete manner.

The minimal variation across socioeconomic tertiles suggests that water access is robust to smaller disruptions, such as the common minor floods (i.e., water-logging). Our findings differed from previous studies, which found that slum residents resorted to open defecation during water logging as latrines became inaccessible [29], and that low socioeconomic status (SES) correlates with reduced resilience of the sanitation infrastructures [3]. There were no extreme weather events in Dhaka in the 12 months before the study period, thus water-logging events in our study were not severe disasters. In that regard, the stability of water and sanitation access during water-logging in our study findings does not imply resilience during catastrophic events or robustness against long-term climate impacts, which severely impact marginalized populations [30].

With regard to access to hygiene facilities, households with basic hygiene during normal periods showed robustness during water-logging periods, whereas some households with limited or no facilities during normal periods had interestingly higher levels of access during water-logging periods. The upward shift could be attributed in part to either relocation [28] or distribution of soap for hand hygiene by civil societies during events [31]. On a related note, hand hygiene behaviors may need improvement. Less than one-quarter of the participants reported always washing their hands with water and soap after toilet use, which implied that more than three-quarters did not. Those who did not always wash their hands during the normal period were unlikely to wash their hands frequently during water-logging periods. There was little upward shift and substantial downward shift (i.e., regression) in hand hygiene behavior during normal vs. water logging periods. The reason for the shift is unclear, but it is likely that household members may have felt the need to conserve water for other purposes during water-logging periods, hence the lower prevalence. The findings highlighted the difference between access to hand hygiene facilities and materials vs. hand hygiene behaviors [32]. Community-level behavioral interventions to promote handwashing [32,33] could offer a potential solution to increase handwashing with soap. The presence of water and soap at the majority of households was nonetheless a positive development, as members of households with available water and soap were more likely to report handwashing with water and soap [34].

## Socioeconomic disparities in WASH access during climate stress

The extent to which differences in WASH access during normal vs. water-logging period varied by socioeconomic status was unclear. Access to water and sanitation was practically universal (as mentioned above), which precluded

assessments in these domains. Those in the second tertile of wealth were most likely to report differences between normal vs. water-logging periods regarding hand hygiene behaviors. We do not have a clear explanation or theory regarding this pattern of variation. However, the prevalence of reporting regular handwashing with water and soap at both normal and water-logging periods was very low, thus we could not rule out chance as the best explanation for the observed heterogeneity. Future studies [35] should consider replication of the current study's findings and use other study methods (e.g., structured observation and in-depth interviews) to understand the mechanism of such variations (if they indeed exist) more comprehensively.

### Strengths and limitations

The strength of this study was that we used validated tools from JMP, thus minimizing issues with using unvalidated instruments and enabling comparison of the findings with those from other countries and settings. Additionally, rapid observation of water and soap availability at participants' primary handwashing stations provided a more comprehensive understanding of hygiene in Korail Slum. However, our study also contained a number of limitations. Firstly, as we collected the study data during daytime, most of our participants were stay-at-home mothers and housewives. Women and caregivers are generally more aware of hygiene than men [36], and our sampling method might have limited the ability to generalize the study findings to male residents of Korail. Secondly, hand hygiene behaviors are known to be over-reported due to social desirability and self-serving tendencies [37]. Thus, the potential for information bias is potentially non-negligible in the study findings. Thirdly, we collected the study data in November 2024. Given the rapid on-going changes in Bangladesh socioeconomic and political condition, the findings of this study may not be generalizable to other periods and settings. Fourthly, we only observed the availability of handwashing materials but not handwashing behavior due to lack of time and resources, which limited the scope of our study findings.

### Conclusion

The study examined access to WASH facilities and behaviors in Dhaka's Korail Slum during normal and water-logging periods, considering socioeconomic variations. We found near-universal access to water and improved sanitation during both normal and water-logging periods. Access to hand hygiene facilities was also stable. However, self-reported handwashing was less common during water-logging periods compared to normal periods, especially among consistent users. There were relatively little and unclear patterns of variation in hand hygiene behavior during normal vs. water-logging periods. The findings suggested that WASH infrastructure in Korail Slum showed robustness to small disruptions such as water-logging. However, the scope of the findings did not include more severe natural disasters. Limited generalizability and potential social desirability bias should also be considered in the interpretation of the study findings.

### Supporting information

**S1 File.  Questionnaire in English.**
(PDF)

**S2 File.  Main data file.**
(CSV)

**S1 Table.  Participants' education, income, occupation, and reported household asset ownerships.**
(DOCX)

**S2 Table.  Principal Component loading of selected characteristics or assets as socioeconomic status indicators.**
(DOCX)

**S3 Table. Characteristics and asset ownership of participants by socioeconomic tertiles.**
(DOCX)

**S4 Table. Access to water according to the JMP Service Ladder among participating Korail Slum residents during normal vs. water-logging periods (overall and stratified by socioeconomic tertile).**
(DOCX)

**S5 Table. Access to sanitation facilities according to the JMP Service Ladder among participating Korail Slum residents during normal vs. water-logging periods (overall and stratified by socioeconomic tertile).**
(DOCX)

**S6 Table. Access to hygiene facilities according to the JMP Service Ladder and self-reported frequency of hand-washing behaviors among participating Korail Slum residents during normal vs. water-logging periods (overall and stratified by socioeconomic tertile).**
(DOCX)

## Acknowledgments

We wish to thank all study participants for their valuable time. We also wish to thank our research assistants who tirelessly conducted interviews on KoboToolbox platform. This study was part of the first author's (MMR) thesis, which was completed in partial fulfillment of the requirements for a Master of Science (M.Sc) degree in Epidemiology at Prince of Songkla University, Hat Yai, Songkhla, Thailand.

## Author contributions

**Conceptualization:** Virasakdi Chongsuvivatwong, Wit Wichaidit.

**Data curation:** Md Mostafizur Rahman.

**Formal analysis:** Md Mostafizur Rahman, Wit Wichaidit.

**Funding acquisition:** Virasakdi Chongsuvivatwong.

**Investigation:** Md Mostafizur Rahman, Wit Wichaidit.

**Methodology:** Md Mostafizur Rahman, Wit Wichaidit.

**Project administration:** Md Mostafizur Rahman, Md Shamim Hayder Talukdar.

**Resources:** Virasakdi Chongsuvivatwong, Md Shamim Hayder Talukdar.

**Software:** Virasakdi Chongsuvivatwong.

**Supervision:** Alan F. Geater, Md Shamim Hayder Talukdar, Wit Wichaidit.

**Validation:** Md Mostafizur Rahman, Alan F. Geater, Md Shamim Hayder Talukdar, Wit Wichaidit.

**Writing – original draft:** Md Mostafizur Rahman, Wit Wichaidit.

**Writing – review & editing:** Md Mostafizur Rahman, Virasakdi Chongsuvivatwong, Alan F. Geater, Md Shamim Hayder Talukdar, Wit Wichaidit.

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
