## [Decision Letter · Decision Letter 0]

21 May 2025

PONE-D-25-21002Differences in access to water, sanitation, and hygiene facilities among residents of Korail Slum, Bangladesh, during normal vs. water-logging situationsPLOS ONE

Dear Dr. Wichaidit,

Thank you for submitting your manuscript to PLOS ONE. After careful consideration, we feel that it has merit but does not fully meet PLOS ONE’s publication criteria as it currently stands. Therefore, we invite you to submit a revised version of the manuscript that addresses the points raised during the review process. Both reviewers have provided a lot of detailed comments which need careful addressing.   In particular they have both highlighted the need for a global contextualisation of the research, and also a clear definition of "water logging".

We look forward to receiving your revised manuscript.

Kind regards,

Alison Parker

Academic Editor

PLOS ONE

Journal Requirements:

2. In the ethics statement in the Methods, you have specified that verbal consent was obtained. Please provide additional details regarding how this consent was documented and witnessed, and state whether this was approved by the IRB

“The first author (MMR) received financial support for data collection in this research work from the TUYF Charitable Trust: Research Capacity through Education and Networking on Epidemiology in Asia, the Department of Epidemiology, Faculty of Medicine, Prince of Songkla University (Grant number 1/2023). The funders had no role in study design, data collection and analysis, decision to publish, or preparation of the manuscript.”

“The first author (MMR) received financial support for data collection in this research work from the TUYF Charitable Trust: Research Capacity through Education and Networking on Epidemiology in Asia, the Department of Epidemiology, Faculty of Medicine, Prince of Songkla University (Grant number 1/2023). The funders had no role in study design, data collection and analysis, decision to publish, or preparation of the manuscript.”

Reviewers' comments:

Reviewer's Responses to Questions

**Comments to the Author**

1. Is the manuscript technically sound, and do the data support the conclusions?

Reviewer #1: Partly

Reviewer #2: Partly

2. Has the statistical analysis been performed appropriately and rigorously? 

Reviewer #1: I Don't Know

Reviewer #2: No

3. Have the authors made all data underlying the findings in their manuscript fully available?

Reviewer #1: Yes

Reviewer #2: No

4. Is the manuscript presented in an intelligible fashion and written in standard English?

Reviewer #1: No

Reviewer #2: Yes

5. Review Comments to the Author

Reviewer #1: Dear Author,

This paper examines the impact of waterlogging on slum residents’ handwashing behaviours and access to WASH facilities in Dhaka. However, several areas require adjustment to meet publishing standards, as detailed below:

1. Language and Style: The language used in this paper requires proofreading and polishing to adhere to academic writing standards. The current manuscript contains several grammatical and structural errors that hinder the clear and complete expression of the research findings and their implications.

2. Introduction: On Introduction page two, lines 1-3, the study focuses on slum residents, while the term 'wealth' suggests a broader scope. Please clarify why data collected from slum residents can be used to analyse 'disparities in these differences by wealth' and provide more detail on how this data contributes to the analysis.

3. Methods: On Methods page one, line 7, please provide a source for the population data.

4. Methods: On Methods page one, line 9, please include a figure to illustrate the location and spatial characteristics of the slum.

5. Sampling Method: On the sampling method page, line 7, please provide further details on the origin of the number '16536,' as this is the first instance of its appearance.

6. Study Instruments: On the study instruments page, line 15, the term 'smartphone-based face-to-face structured interview questionnaire' may benefit from rephrasing for improved clarity.

7. Study Instruments: On the study instruments page, line 18, it would be beneficial to include citations for the 'previous studies' mentioned.

8. Study Variables: On the study variables page, lines 3-5, 'waterlogging' is presented as a general term representing varying degrees of reduced access to WASH facilities due to surface water. Please specify the quality control measures employed to ensure consistency in how interviewees perceived and reported 'waterlogging.'

9. Study Variables: On the study variables page, line 21, provide more detailed information about 'the most recent water-logging period,' including its scale and severity (e.g., water depth). This will clarify whether all interviewees referred to the same event or events with comparable impacts.

10. Effect Modifier Variable: On the effect modifier variable page, line 11, please elaborate on the 'loading scores' method used to categorize participants into tertiles.

11. Results: On the results page, line 8, please provide context for the income figure of 20,000 Taka per household per month by indicating its relative level within the Bangladeshi income distribution.

12. Discussion: On the discussion page, line 20, if the reference pertains to open defecation during 'flooding,' clarify the relationship between 'waterlogging' and 'flooding' in the definition of waterlogging (i.e., whether waterlogging represents minor flooding, and how ‘minor’ it is).

13. Discussion: The discussion section would benefit from addressing some of the interesting results presented. For example, Table 2 shows that 299 households had access to limited sanitation facilities under normal conditions, but 13 of these households reported access to basic sanitation facilities during waterlogging. Please provide a discussion of this finding.

14. Discussion: Consider adding a discussion on the potential resilience of basic and limited WASH facilities to waterlogging, and how this might offer an alternative interpretation of the results.

15. Results and Discussion: In addition to the reported frequencies of handwashing during normal and waterlogging periods, it would be valuable to include the frequencies of access to WASH facilities during these periods. For instance, while 76 out of 84 households with access to basic sanitation facilities in normal periods retained access during waterlogging, please discuss whether their usage frequency was affected. The questionnaire appears to contain relevant and valuable data and expanding the discussion of this information would enhance the paper's research significance.

16. Presentation: In Tables 2-5, consider improving the presentation of the data. For example, in Table 2, ensure consistency in the formatting (e.g., font size, background colour) of 'Basic' in both the vertical and horizontal axes.

17. Table 5: Table 5 requires further clarification. For instance, for the 116 households reported as not always having access to hand hygiene facilities during normal periods, please specify whether this refers to access to basic or limited facilities.

Addressing these points will significantly strengthen the manuscript.

Reviewer #2: Reviewer Comments on Manuscript PONE-D-25-21002

This manuscript examines differences in access to water, sanitation, and hygiene (WASH) services among residents of Korail Slum, Dhaka, Bangladesh during normal and water-logged conditions. The topic is important and underrepresented in current literature, particularly in relation to urban slums in South Asia, and the authors have access to a valuable dataset. However, in its current form, the paper requires substantial revisions to reach publishable quality.

Major Concerns:

The manuscript is a lack of contextualisation for an international audience. Key concepts such as “water-logging” are not clearly defined or situated within the broader literature. The structure is also problematic, with a particularly weak discussion section that does not clearly address the stated objectives or elaborate on the implications of the findings. Additionally, there are no visual representations of data, which limits reader to understand the geographical and hydrological context of the study. The results section is brief and lacks depth, while the interpretation remains superficial. The paper also fails to engage with relevant recent literature on WASH infrastructure, urban flood resilience, and disaster-related behaviour change. These shortcomings collectively weaken the paper’s potential to contribute meaningfully to global scholarship.

Specific Comments:

• Currently it lack international context. International readers would benefit from a clearer explanation of the urban flooding phenomenon, how it manifests in Korail Slum, and its relation to broader global challenges in slum-based WASH resilience.

• Clarify or remove the discussion of “paddy flooding” in the introduction—it is not currently connected to the context.

• Provide a clear rationale for choosing Korail Slum, emphasizing its unique environmental and demographic challenges.

• Add a detailed map of the study area.

• The manuscript includes a large number of tables, but the overall data visualisation could be improved. Consider summarising key findings graphically (e.g. through charts or graphs) to enhance clarity and reader engagement. The statistics provided in the tables can go to the supplementary.

• The discussion section needs clearer subheadings and should directly address the aim and objectives with interpretation.

• In discussion, clarify the logic behind behaviour regression—for example, why those who report handwashing during normal periods did not maintain the behaviour during water-logging.

• Address the unexplained trend where the middle socioeconomic tertile shows the greatest shift in hygiene behaviour.

• Polish the language throughout the manuscript to improve readability and clarity.

6. PLOS authors have the option to publish the peer review history of their article (what does this mean? ). If published, this will include your full peer review and any attached files.

**Do you want your identity to be public for this peer review?** For information about this choice, including consent withdrawal, please see our Privacy Policy .

Reviewer #1: **Yes: ** Zhe Zhan

Reviewer #2: No

---

## [Author Response · Author response to Decision Letter 1]

4 Jul 2025

Editor’s Comment / Journal Requirements

COMMENT

https://journals.plos.org/plosone/s/file?id=wjVg/

PLOSOne_formatting_sample_main_body.pdf and

https://journals.plos.org/plosone/s/file?id=ba62/

PLOSOne_formatting_sample_title_authors_affiliations.pdf

RESPONSE

We thank the editor for the comment. We have revised the manuscript accordingly.

COMMENT

2. In the ethics statement in the Methods, you have specified that verbal consent was obtained. Please provide additional details regarding how this consent was documented and witnessed, and state whether this was approved by the IRB

RESPONSE

Thank you for this important comment. We have revised the Ethics section in the Methods to clarify the procedures for verbal consent. Specifically, we have added the following information:

“We requested and received ethical approval for this study from the Bangladesh Medical Research Council (BMRC) (Ref: BMRC/NREC/2022-2025/561). Due to the low literacy levels among the residents of Korail Slum, we requested and received approval for a waiver of written informed consent and the use of only verbal informed consent.

We obtained verbal informed consent from all participants before initiating data collection. This process included reading a consent script aloud. The consent script included a summary of the purpose of the study, procedures, risks, benefits, and confidentiality. We gave the participants an opportunity to ask questions before verbally agreeing to participate or declining participation. Our research assistants documented the verbal consent directly in the electronic questionnaire form on KoboCollect. The form required an affirmative response regarding verbal consent to display the data collection questionnaire.”

COMMENT

“The first author (MMR) received financial support for data collection in this research work from the TUYF Charitable Trust: Research Capacity through Education and Networking on Epidemiology in Asia, the Department of Epidemiology, Faculty of Medicine, Prince of Songkla University (Grant number 1/2023). The funders had no role in study design, data collection and analysis, decision to publish, or preparation of the manuscript.”

“The first author (MMR) received financial support for data collection in this research work from the TUYF Charitable Trust: Research Capacity through Education and Networking on Epidemiology in Asia, the Department of Epidemiology, Faculty of Medicine, Prince of Songkla University (Grant number 1/2023). The funders had no role in study design, data collection and analysis, decision to publish, or preparation of the manuscript.”

RESPONSE

Thank you for your guidance. We have removed the funding-related text from the manuscript. We wish to maintain the same funding statement as it appears, but we will leave the decision on where to place the statement at the editors’ discretion. We have included the following remarks in the cover letter for this revision.

“We wish to take this opportunity to request the Journal to state the following remarks in the Funding Statement, should the manuscript be accepted for publication by PLoS ONE:

“The first author (MMR) received financial support for data collection in this research work from the TUYF Charitable Trust: Research Capacity through Education and Networking on Epidemiology in Asia, the Department of Epidemiology, Faculty of Medicine, Prince of Songkla University (Grant number 1/2023). The funders had no role in study design, data collection and analysis, decision to publish, or preparation of the manuscript.”

COMMENT

RESPONSE

We thank the Editor for the comments. We have included captions for the Supporting Information files at the end of the manuscript accordingly.

Reviewers' comments:

Reviewer #1:

COMMENT

Dear Author,

This paper examines the impact of waterlogging on slum residents’ handwashing behaviours and access to WASH facilities in Dhaka. However, several areas require adjustment to meet publishing standards, as detailed below:

RESPONSE

Thank you for your valuable comments and suggestions to improve the clarity, quality, and scientific rigor of our manuscript. We have tried to revise the manuscript and address each point raised to the extent possible.

COMMENT

1. Language and Style: The language used in this paper requires proofreading and polishing to adhere to academic writing standards. The current manuscript contains several grammatical and structural errors that hinder the clear and complete expression of the research findings and their implications.

RESPONSE

We appreciate this feedback. We have thoroughly proofread and revised the entire manuscript for language, grammar, and clarity to ensure adherence to academic writing standards.

COMMENT

2. Introduction: On Introduction page two, lines 1-3, the study focuses on slum residents, while the term 'wealth' suggests a broader scope. Please clarify why data collected from slum residents can be used to analyse 'disparities in these differences by wealth' and provide more detail on how this data contributes to the analysis.

RESPONSE

Thank you for pointing this out. We just want to clarify that we did not select slum residents specifically for the study of socioeconomic disparities. We measured WASH access among slum residents due to their relative vulnerability (particularly during water logging periods compared to normal periods), and we needed to measure the extent to which socioeconomic disparities occurred in this context. The idea behind this assessment was that even among those who lived in the slums, there was heterogeneity regarding socioeconomic conditions. In various surveys (such as the MICS) asset ownership is used as a proxy indicator of socioeconomic status. We used the same approach in this study. We have revised the INTRODUCTION section as follows:

“Although slum residents are generally less wealthy than residents of other parts of a city, heterogeneity in socioeconomic conditions (i.e., wealth disparities) can also be found within a slum and can be measured using a combination of socioeconomic characteristics and asset ownership. Despite concerns regarding the vulnerability of slum residents to climate events and the general notion that climate events affect the poor more commonly and to a greater extent compared to the rich, studies have not described differences in access to water, sanitation, and hygiene (WASH) facilities within a household between normal and water-logging periods among slum residents. Studies also have not described the disparities of these differences by wealth. ”

COMMENT

3. Methods: On Methods page one, line 7, please provide a source for the population data.

RESPONSE

We thank the reviewer for the comment. We have added details to the Sampling Method sub-section with citations accordingly.

COMMENT

4. Methods: On Methods page one, line 9, please include a figure to illustrate the location and spatial characteristics of the slum. RESPONSE

We thank the reviewer for the suggestion. We have added Figure 1, which includes a map of Korail Slum with the location in Dhaka and proximity to key city landmarks.

COMMENT

5. Sampling Method: On the sampling method page, line 7, please provide further details on the origin of the number '16536,' as this is the first instance of its appearance.

RESPONSE

We thank the reviewer for the comment. We have added details to the Sampling Method sub-section with citations accordingly.

COMMENT

6. Study Instruments: On the study instruments page, line 15, the term 'smartphone-based face-to-face structured interview questionnaire' may benefit from rephrasing for improved clarity.

RESPONSE

We have revised the remarks to the following to improve clarity:

“Our study instrument was a structured questionnaire hosted on the web-based KoboToolbox platform. We used smartphones with either a web browser or the KoboCollect application for data collection.”

COMMENT

7. Study Instruments: On the study instruments page, line 18, it would be beneficial to include citations for the 'previous studies' mentioned.

RESPONSE

We thank the reviewer for the suggestion. We have added the citations accordingly.

COMMENT

8. Study Variables: On the study variables page, lines 3-5, 'waterlogging' is presented as a general term representing varying degrees of reduced access to WASH facilities due to surface water. Please specify the quality control measures employed to ensure consistency in how interviewees perceived and reported 'waterlogging.'

RESPONSE

We thank the reviewer for the comment. We have added the following remarks to the Study Variables sub-section in METHODS:

"Our operational definition of the term "water-logging" was "inundation of least 25 cm in depth lasting 10 hours or more." We trained our research assistants to explain this definition in a uniform manner to all participants when conducting interviews about experiences during water-logging periods."

COMMENT

9. Study Variables: On the study variables page, line 21, provide more detailed information about 'the most recent water-logging period,' including its scale and severity (e.g., water depth). This will clarify whether all interviewees referred to the same event or events with comparable impacts.

RESPONSE

We thank the reviewer for the comment. We have added the following remarks to the Study Variables sub-section in METHODS:

"We asked participants to describe their level of access to water, sanitation, and hygiene during the most recent water-logging periods within the past 12 months that the participants experienced without giving the specific dates."

COMMENT

10. Effect Modifier Variable: On the effect modifier variable page, line 11, please elaborate on the 'loading scores' method used to categorize participants into tertiles.

RESPONSE

We thank the reviewer for the suggestion. We made a mistake in writing. We did not use the loading to rank the participants, we used the PCA scores. We apologize for our mistake and have made corrections in the manuscript texts accordingly as follows:

“The investigators then followed the instructions for Principal Component Analysis (PCA) and obtained the factor scores. We then performed descriptive statistical analysis on the factor scores and used the 33rd and 67th percentiles as the cut-off points for the socioeconomic tertiles.”

COMMENT

11. Results: On the results page, line 8, please provide context for the income figure of 20,000 Taka per household per month by indicating its relative level within the Bangladeshi income distribution.

RESPONSE

We thank the reviewer for the comment. Although the exact percentile could not be determined based on existing data, the household income of 20,000 BDT per month is approximately 45 percent of the average income in urban areas and 77 percent of the average income in rural areas in 2022 according to the Bureau of Statistics

Citation: https://bbs.portal.gov.bd/sites/default/files/files/bbs.portal.gov.bd/page/57def76a_aa3c_46e3_9f80_53732eb94a83/2023-04-13-09-35-ee41d2a35dcc47a94a595c88328458f4.pdf

We feel that based on the need to mention statistics, the addition of the context would be more appropriate elsewhere. Thus, we added the following remarks in the METHODS section.

""The household income of 20,000 BDT per month was approximately half of the average household income in urban areas of Bangladesh in 2022, and we used the amount as a proxy cut-off point for being low-income in the slum setting.""

COMMENT

12. Discussion: On the discussion page, line 20, if the reference pertains to open defecation during 'flooding,' clarify the relationship between 'waterlogging' and 'flooding' in the definition of waterlogging (i.e., whether waterlogging represents minor flooding, and how ‘minor’ it is).

RESPONSE

Thank you for your comment. We have re-checked the reference document, and revised the remark in the DISCUSSION section from the following:

"This contrasted starkly with findings from 2000 to 2010 that open defecation was common in urban slums, particularly during flooding [Reference]. "

To the following:

"The absence of open defecation suggested that the prevalence of open defecation in Bangladesh continued to decline after the reduction to one percent in 2015[Reference]"

COMMENT

13. Discussion: The discussion section would benefit from addressing some of the interesting results presented. For example, Table 2 shows that 299 households had access to limited sanitation facilities under normal conditions, but 13 of these households reported access to basic sanitation facilities during waterlogging. Please provide a discussion of this finding.

RESPONSE

We appreciate the reviewer’s suggestion. We have added the following remarks in the DISCUSSION section:

"The majority of households had limited sanitation during normal periods due to the act of sharing their improved latrine with people from other households. However, a small minority (approximately four percent) of participants from these households reported having access to basic (non-shared) sanitation facilities during water-logging periods. We do not know what accounted for this shift, although a previous study reported that slum residents would relocate housing within the same area during water-logging, such as with family members, and relocated residence might have non-shared toilet. Future studies should consider asking questions regarding temporary relocation during water-logging periods."

COMMENT

14. Discussion: Consider adding a discussion on the potential resilience of basic and limited WASH facilities to waterlogging, and how this might offer an alternative interpretation of the results.

RESPONSE

We thank the reviewer for the comment. We actually considered the more dynamic shifts in hygiene access and included the following remarks in the DISCUSSION section:

"With regard to access to hygiene facilities, households with basic hygiene during normal periods showed robustness during water-logging periods, whereas some households with limited or no facilities during normal periods had interestingly higher levels of access during water-logging periods. The upward shift could be attributed in part

---

## [Decision Letter · Decision Letter 1]

29 Jul 2025

PONE-D-25-21002R1Differences in access to water, sanitation, and hygiene facilities among residents of Korail Slum, Bangladesh, during normal vs. water-logging situationsPLOS ONE

Dear Dr. Wichaidit,

Thank you for submitting your manuscript to PLOS ONE. After careful consideration, we feel that it has merit but does not fully meet PLOS ONE’s publication criteria as it currently stands. Therefore, we invite you to submit a revised version of the manuscript that addresses the points raised during the review process.

Both reviewers are proposing further minor changes.

We look forward to receiving your revised manuscript.

Kind regards,

Alison Parker

Academic Editor

PLOS ONE

Journal Requirements:

Reviewers' comments:

Reviewer's Responses to Questions

**Comments to the Author**

1. If the authors have adequately addressed your comments raised in a previous round of review and you feel that this manuscript is now acceptable for publication, you may indicate that here to bypass the “Comments to the Author” section, enter your conflict of interest statement in the “Confidential to Editor” section, and submit your "Accept" recommendation.

Reviewer #1: All comments have been addressed

Reviewer #2: (No Response)

2. Is the manuscript technically sound, and do the data support the conclusions?

Reviewer #1: Yes

Reviewer #2: Yes

3. Has the statistical analysis been performed appropriately and rigorously? 

Reviewer #1: I Don't Know

Reviewer #2: No

4. Have the authors made all data underlying the findings in their manuscript fully available?

Reviewer #1: Yes

Reviewer #2: No

5. Is the manuscript presented in an intelligible fashion and written in standard English?

Reviewer #1: No

Reviewer #2: Yes

6. Review Comments to the Author

Reviewer #1: Thank you for resubmitting your work and for incorporating the previous feedback. However, I still saw several areas that could benefit from further improvement:

Introduction, line 50, ‘floods are very common…’: please rephrase so readers can get more information about the floods in Bangladesh, i.e. frequency, severity, damage, etc.

Introduction, line 75, ‘water, sanitation, and hygiene (WASH): please use WASH from here as you already mentioned the full term in the previous text.

Introduction, line 77, ‘such data are…’: please highlight your research significance and be more specific - what benefits of knowing the disparities of these differences by wealth?

Methods, line 89, ‘Figure 1’: It's good to see that you've added a figure here, but could you replace it with figures from a different source, such as government documents or other literature? I personally am not inclined to use a screenshot from Google satellite in my paper, but this is just a personal preference rather than a comment.

Methods, line 90, ‘151,500 people’: please add references here.

Methods, line 97, ‘we excluded…’: more details needed here on how you excluded these potential participants - did you ask their health conditions, or based on the observations only?

Methods, line 126, ‘the lead author…’: please indicate who compared and checked the discrepancies? The lead author or 'we'?

Methods, line 147, ‘surface water’: lower case ‘s’

Methods, line 156, ‘captured information…’: please indicate if these data were analysed or included in this paper

Methods, line 180, ‘key moments…’: please indicate what 'key moments' are. More information needed here.

Methods, line 198, ‘(PCA)’: Move (PCA) before the reference, maybe?

Methods, line 223, ‘data-related issues’: please indicate what data-related issues are.

Results, line 275, ‘however, the difference.’: you may want to rephrase this sentence as you already used 'However' in the last sentence.

Discussion, line 327, ‘as residents shifted…’: More details needed here - was this derived from the data collected? If so, it is worth referring to it.

Discussion, line 329, ‘open defecation in…’: Here you used 'contradicted', could you be more specific on this argument here - in reference 28, people chose open defecation because they were not willing to use a shared pit latrine? Or because there was no pit latrine available at all? Are you making an argument that residents in Korail slum were still willing to share toilets with other people, which is different from the Uganda case?

Discussion, line 330, ‘corelates with…’: could you be more specific on 'reduced resilience of the sanitation infrastructures' as the original source was mainly about the diarrhoea cases.

Discussion, line 359, ‘a previous study…’: More details needed here - what did this previous study find?

Table 4, line 555: Should the title be 'access to sanitation'?

Table 4, line 558: superscript 'b' was not in Table 4

Table 5, should ‘0.023’ be non-bold?

I strongly suggest giving your manuscript another thorough proofreading, as there are still some unresolved structural and grammatical issues I haven't detailed.

Reviewer #2: Reviewer Comments

The revised manuscript presents meaningful improvements over the original submission and demonstrates the authors’ effort to address many of the concerns raised in the initial review. The authors revised the manuscript's structure, clarified terminology (such as “water-logging”), and improved contextual framing for a broader audience. The addition of a map (Figure 1) is an enhancement and helps to visualise the study area geographically.

The manuscript now includes clearer subheadings and demonstrates better alignment between objectives, results, and discussion. The clarity have improved meaningfully. However, several areas would benefit from additional refinement before publication:

• Data Visualization: I am not agreeing with the authors here, while the number of tables remains high, no figures summarising key trends or regression results. The rational about reproducibility around table is not valid as there is no survey data given that one can use to reproduce the tables.

Even simple bar charts or summary visuals would help general readers to interpret findings. Consider moving some tables to supplementary material and adding at least few summary figures.

For example, Tables 2 and 5 could be combine into a single panel figure (e.g., side-by-side bar plots showing access and behaviour across SES groups under normal vs. waterlogged conditions).

Also, authors may use difference plots (e.g., percent change between normal and waterlogged periods by SES) to visually represent which groups were most impacted.

The current satellite map of the study area (Korail Basti) lacks sufficient annotation to be informative and accessible to an international audience unfamiliar with the local geography. Including an inset map for geographic context, along with clearly marked survey locations, would significantly enhance the clarity and interpretability of the figure.

• Discussion: The discussion could interpret and contextualize more clearly under the appropriate subheadings addressing the aim and objectives of the paper. These might be like

o Changes in WASH access and behaviours during water-logging

o Socioeconomic disparities in WASH access during climate stress

o Implications for public health and urban resilience

7. PLOS authors have the option to publish the peer review history of their article (what does this mean? ). If published, this will include your full peer review and any attached files.

**Do you want your identity to be public for this peer review?** For information about this choice, including consent withdrawal, please see our Privacy Policy .

Reviewer #1: **Yes: ** Zhe Zhan

Reviewer #2: No

---

## [Author Response · Author response to Decision Letter 2]

26 Aug 2025

Response to Reviewers Comments

Reviewer #1: Reviewer Comments

COMMENT

Thank you for resubmitting your work and for incorporating the previous feedback. However, I still saw several areas that could benefit from further improvement:

Introduction, line 50, ‘floods are very common…’: please rephrase so readers can get more information about the floods in Bangladesh, i.e. frequency, severity, damage, etc.

Response

Thank you for the suggestion. We have revised the sentence to read:

“Bangladesh experiences an average of two to five major floods per year, which can inundate up to two-thirds of the country and cause severe damage to infrastructure, agriculture, and public health systems.”

COMMENT

Introduction, line 75, ‘water, sanitation, and hygiene (WASH): please use WASH from here as you already mentioned the full term in the previous text.

Response

Thank you. We have replaced “water, sanitation, and hygiene (WASH)” with simply “WASH” throughout the remainder of the manuscript.

COMMENT

Introduction, line 77, ‘such data are…’: please highlight your research significance and be more specific - what benefits of knowing the disparities of these differences by wealth?

Response: We have revised the sentence to clarify the significance as follows:

“Such data can inform targeted interventions by helping public health and urban planning stakeholders identify population groups that are vulnerable to WASH disruptions during climate-related events.”

COMMENT

Methods, line 89, ‘Figure 1’: It's good to see that you've added a figure here, but could you replace it with figures from a different source, such as government documents or other literature? I personally am not inclined to use a screenshot from Google satellite in my paper, but this is just a personal preference rather than a comment.

Response: Thank you for the suggestion. The authors have deliberated and decided to keep the figure as it appears. We have made some changes as per the suggestions of another reviewer

COMMENT

Methods, line 90, ‘151,500 people’: please add references here.

Response: We thank the reviewer for the comment. We have added one reference accordingly.

COMMENT

Methods, line 97, ‘we excluded…’: more details needed here on how you excluded these potential participants - did you ask their health conditions, or based on the observations only?

Response: Thank you for your question. We have revised the sentence to clarify the procedure:

“We excluded individuals who self-reported illnesses or other impairments that they deemed would prevent them from participating fully in the interview.”

COMMENT

Methods, line 126, ‘the lead author…’: please indicate who compared and checked the discrepancies? The lead author or 'we'?

Response: We have revised the sentence to the following:

“The lead author (MR) compared the original and the back-translated English versions, identified discrepancies, and made corrections to the corresponding parts of the Bengali translation.”

COMMENT

Methods, line 147, ‘surface water’: lower case ‘s’

Response: Thank you for noticing. We have corrected it to “surface water,”

COMMENT

Methods, line 156, ‘captured information…’: please indicate if these data were analysed or included in this paper

Response: We have revised the sentence to:

“If a participant switched to a different source during water logging, we would ask the participant to identify the source using the same answer choices as our measurement of access to water during normal periods. We also asked the participants about changes in collection time, water shortages, and treatment practices.”

COMMENT

Methods, line 180, ‘key moments…’: please indicate what 'key moments' are. More information needed here.

Response: We have added the following clarification:

“Key moments for hand hygiene referred to events identified by the WHO/UNICEF Joint Monitoring Programme[23] as moments where hand hygiene is essential to prevent disease transmission, namely: after going to the toilet, after changing a child's soiled diaper, before preparing food, before feeding a child, and before eating.”

COMMENT

Methods, line 198, ‘(PCA)’: Move (PCA) before the reference, maybe?

Response: We agree with the suggestion and have revised the sentence accordingly so that (PCA) appears before the reference citation.

COMMENT

Methods, line 223, ‘data-related issues’: please indicate what data-related issues are.

Response: We have added the following remarks after the term “data-related issues”:

“(e.g., missing values, duplicated entries)”

COMMENT

Results, line 275, ‘however, the difference.’: you may want to rephrase this sentence as you already used 'However' in the last sentence.

Response: We thank the reviewer for the comment. The RESULTS section has been extensively revised. We decided to restructure the results tables and move them to the supplementary section (as per the suggestions of another reviewer), and added figures.

COMMENT

Discussion, line 327, ‘as residents shifted…’: More details needed here - was this derived from the data collected? If so, it is worth referring to it.

Response: We thank the reviewer for the comment. We decided that the remark was not needed to convey the main idea in the paragraph and decided to delete the sentence.

COMMENT

Discussion, line 329, ‘open defecation in…’: Here you used 'contradicted', could you be more specific on this argument here - in reference 28, people chose open defecation because they were not willing to use a shared pit latrine? Or because there was no pit latrine available at all? Are you making an argument that residents in Korail slum were still willing to share toilets with other people, which is different from the Uganda case?

Response: We thank the reviewer for the comment. We have revised the remarks as follow:

“Our findings differed from previous studies, which found that slum residents resorted to open defecation during water logging as latrines became inaccessible[29], and that low socioeconomic status (SES) correlates with reduced resilience of the sanitation infrastructures[3].”

COMMENT

Discussion, line 330, ‘corelates with…’: could you be more specific on 'reduced resilience of the sanitation infrastructures' as the original source was mainly about the diarrhoea cases.

Response: Thank you for the comment. We have removed the citation and made the following changes to the remark:

“Our findings differed from previous studies, which found that slum residents resorted to open defecation during water logging as latrines became inaccessible[29], and that low socioeconomic status (SES) correlates with reduced resilience of the sanitation infrastructures[3].”

COMMENT

Discussion, line 359, ‘a previous study…’: More details needed here - what did this previous study find?

Response: We thank the reviewer for the comment. We have deleted the sentence in its entirety.

COMMENT

Table 4, line 555: Should the title be 'access to sanitation'?

Response: We thank the reviewer for the comment, and we agree. The title has been changed to “JMP Service ladder access to sanitation facilities” accordingly. However, as the other reviewer suggest that we use graphs instead of tables, the current Tables 2 thru Table 5 have been moved to the supplementary information section as Supplementary Tables 4 thru 6. We nonetheless thank the reviewer for the helpful suggestion and for the attention to the presentation of our study findings.

COMMENT

Table 4, line 558: superscript 'b' was not in Table 4

Response: We appreciate the careful review. We have revised Table 4 and added the missing superscript ‘b’ where appropriate.

COMMENT

Table 5, should ‘0.023’ be non-bold?

Response: Yes, we have corrected the formatting and removed the bold typeface from Table 5.

COMMENT

I strongly suggest giving your manuscript another thorough proofreading, as there are still some unresolved structural and grammatical issues I haven't detailed.

Response: We sincerely thank the reviewer for this recommendation. We have carefully proofread the manuscript again and addressed all remaining grammatical and structural inconsistencies to improve clarity and readability.

###################################################

#### End of Comments from Reviewer #1 ####

###################################################

Reviewer #2: Reviewer Comments

COMMENT

The revised manuscript presents meaningful improvements over the original submission and demonstrates the authors’ effort to address many of the concerns raised in the initial review. The authors revised the manuscript's structure, clarified terminology (such as “water-logging”), and improved contextual framing for a broader audience. The addition of a map (Figure 1) is an enhancement and helps to visualise the study area geographically.

The manuscript now includes clearer subheadings and demonstrates better alignment between objectives, results, and discussion. The clarity have improved meaningfully. However, several areas would benefit from additional refinement before publication:

Response: We sincerely thank the reviewer for acknowledging the improvements in our revised manuscript. We appreciate the continued constructive feedback and have further revised the manuscript to enhance clarity, structure, and presentation of findings, as detailed below.

COMMENT

Data Visualization: I am not agreeing with the authors here, while the number of tables remains high, no figures summarising key trends or regression results. The rational about reproducibility around table is not valid as there is no survey data given that one can use to reproduce the tables.

Even simple bar charts or summary visuals would help general readers to interpret findings. Consider moving some tables to supplementary material and adding at least few summary figures.

For example, Tables 2 and 5 could be combine into a single panel figure (e.g., side-by-side bar plots showing access and behaviour across SES groups under normal vs. waterlogged conditions).

Also, authors may use difference plots (e.g., percent change between normal and waterlogged periods by SES) to visually represent which groups were most impacted.

The current satellite map of the study area (Korail Basti) lacks sufficient annotation to be informative and accessible to an international audience unfamiliar with the local geography. Including an inset map for geographic context, along with clearly marked survey locations, would significantly enhance the clarity and interpretability of the figure.

RESPONSE:

We thank the reviewer for the comment. We have revised the map in Figure 1 accordingly

With regard to the presentation of the study findings, the authors deliberated and decided that the alluvial plot would be the best way to visualize our study findings. We have modified the contents of the Tables 2 thru 5 and moved them to the Supplementary Information section (as Supplementary Tables 4 thru 6). We have also added Figures 2 thru 4 (alluvial plots) to the RESULTS section accordingly. We have also extensively revised the RESULTS section to suit the restructuring of the results tables and the figures.

COMMENT

Discussion: The discussion could interpret and contextualize more clearly under the appropriate subheadings addressing the aim and objectives of the paper. These might be like

o Changes in WASH access and behaviours during water-logging

o Socioeconomic disparities in WASH access during climate stress

o Implications for public health and urban resilience

RESPONSE:

After internal deliberations, we decided to add the following sub-headers to the DISCUSSION section:

o Changes in WASH access and behaviours during water-logging

o Socioeconomic disparities in WASH access during climate stress

o Strengths and Limitations

---

## [Editor Report · Decision Letter 2]

1 Sep 2025

Differences in access to water, sanitation, and hygiene facilities among residents of Korail Slum, Bangladesh, during normal vs. water-logging situations

PONE-D-25-21002R2

Dear Dr. Wichaidit,

We’re pleased to inform you that your manuscript has been judged scientifically suitable for publication and will be formally accepted for publication once it meets all outstanding technical requirements.

Kind regards,

Alison Parker

Academic Editor

PLOS ONE
---

## [Editor Report · Acceptance letter]

PONE-D-25-21002R2

PLOS ONE

Dear Dr. Wichaidit,

I'm pleased to inform you that your manuscript has been deemed suitable for publication in PLOS ONE. Congratulations! Your manuscript is now being handed over to our production team.

Kind regards,

on behalf of

Dr. Alison Parker

Academic Editor

PLOS ONE